# Translation, Adaptation, and Validation of the Modified Thai Version of Champion’s Health Belief Model Scale (MT-CHBMS)

**DOI:** 10.3390/healthcare11010128

**Published:** 2022-12-31

**Authors:** Patinya Suriyong, Surin Jiraniramai, Nahathai Wongpakaran, Kanokporn Pinyopornpanish, Chaisiri Angkurawaranon, Wichuda Jiraporncharoen, Victoria L. Champion, Tinakon Wongpakaran

**Affiliations:** 1Department of Family Medicine, Faculty of Medicine, Chiang Mai University, Chiang Mai 50200, Thailand; 2Department of Psychiatry, Faculty of Medicine, Chiang Mai University, Chiang Mai 50200, Thailand; 3School of Nursing, Indiana University, Indianapolis, IN 46202, USA; 4Melvin and Bren Simon Comprehensive Cancer Center, Indiana University, Indianapolis, IN 46202, USA

**Keywords:** breast cancer, breast cancer screening, breast self-examination, mammogram, breast ultrasound, champion health belief model, health belief model, confirmatory factor analysis, factor structure

## Abstract

Background: While breast cancer is the leading cause of cancer death among Thai women, breast self-examination (BSE), mammography, and ultrasound use are still underutilized. There is a need to assess women’s beliefs about breast cancer and screening in different cultural settings. As a result, a tool to measure the beliefs that influence breast-cancer-screening practices is needed. Champion’s Health Belief Model Scale (CHBMS) is a valid and reliable tool for assessing individuals’ attitudes toward breast cancer and screening methods, but it has not been validated in Thai women. The study aimed to translate and validate the CHBMS for breast self-examination and mammography among Thai women and to modify the original scale by adding ultrasound items for breast cancer screening. In addition, the purpose of this study was to create a modified Thai version of the CHBMS which could be used to better understand patients’ beliefs regarding breast cancer screening in Thailand, in order to develop practical and effective interventions suited to their beliefs. Methods: The CHBMS was translated into Thai, validated by a panel of experts, back-translated, modified by adding content about ultrasound for screening breast cancer, and pretested. Confirmatory factor analysis was used with a sample of 130 Thai women aged 40 to 70 years old. Result: The final MT-CHBMS consisted of 64 items determining ten subscales: susceptibility, seriousness, benefits—breast self-examination, benefits—mammogram, barriers—BSE, barriers—mammogram, confidence, health motivation, benefits—ultrasound, and barriers—ultrasound. The MT-CHBMS demonstrated excellent internal consistency. The ten-factor model was best fitted to the data. Conclusion: The MT-CHBMS was found to be a reliable and valid tool for measuring individuals’ attitudes toward breast cancer and screening methods. The scale could be easily used by healthcare providers to determine the beliefs before planning appropriate interventions to increase early detection.

## 1. Introduction

Breast cancer is the most common malignancy worldwide [1]. In the USA in 2021, it was estimated that 284,200 new cases of invasive breast cancer would be diagnosed in women, with 44,130 deaths yearly. Breast cancer is the most common cancer and the leading cause of female mortality in Thailand [2], although current treatments can help patients live longer. Evidence from the American Cancer Society shows that breast cancer has a good prognosis when detected early [3]. In the non-metastatic stage, the 5-year survival rate is about 99%, while in the metastatic stage, it is only 28%. Early breast cancer often causes no signs or symptoms and is usually diagnosed through mammography screening [2,4].

According to the current American Cancer Society Guidelines for the Early Detection of Cancer 2020 [5], it is recommended that all women over 40 should have a routine mammogram screening for breast cancer every 1–2 years, and recommendations for breast self-examination are not provided. In Thailand, the current guidelines for breast cancer screening [6] include breast cancer screening according to age. For ages 20–39 years old, it is recommended that breast self-examination should be performed once a month. Women between 40 and 69 years should be examined by a doctor annually. If abnormalities are identified, a mammogram will be scheduled. For the age of 70 years old and over, mammography for breast cancer screening should be weighed in terms of benefits and risks based on individual’s life expectancy and preference. However, in the voluntary case of populations who wish to have breast cancer screening by mammogram in the first place, recommendations for screening have been added that are similar to those recommended by the American Cancer Society. This recommendation was caused by public health policy and public finance management in Thailand.

However, despite that recommendation, the breast cancer screening results in Thailand are still low compared to other countries such as the USA. According to the National Institute of Health (NIH) USA in 2018, 72.8% of women aged 50–74 had a mammogram within two years [5], whereas only 10.1% of women aged 30–59 had received a mammogram in Thailand [7]. The main reason provided for not having mammogram was a lack of knowledge about mammograms and a lack of awareness of the need for an examination because they had no symptoms.

In Thailand, breast cancer was discovered in stages 1 and 2 at 33.9%, much lower than the 64% reported in the USA [5,8], implying that fewer Thai women adhered to breast cancer screening guidelines. Lack of knowledge about breast cancer as well as the lack of understanding the importance of screening may contribute to the low number of women screened. Research has underscored the necessity of educating women about proper screening even in the absence of symptoms. Therefore, it is crucial to understand Thai women’s knowledge and beliefs about breast cancer screening to design effective interventions to increase the utilization of screening methods. There is a need to develop a practical evaluation tool to measure individuals’ attitudes toward breast cancer and screening methods in Thailand.

Champion’s Health Belief Model Scale (CHBMS) [9] is one of the widely used screening tools to address the issues concerning the researcher’s question. CHBMS, based on the Health Belief Model, is a tool used to identify knowledge and beliefs influencing health behaviors. The first version of CHBMS was developed to explain breast self-examination (BSE) behavior in the USA [10,11,12,13,14]. In 1993, the confidence subscale for BSE among women was added to the CHBMS [10,11,12,13,14]. The CHBMS was also revised in 1999 to take the increasing use of mammogram screening into account [9].

The scale has four main constructs: susceptibility, seriousness, benefits, and barriers. The constructs are used to predict breast cancer screening behaviors. Health motivation and self-efficacy were added to the original four constructs and was found to be valid and reliable for assessing Western culture’s beliefs about breast cancer and screening methods. The tool has been widely translated and used in many countries, both Western and Eastern cultures [10,11,12,13,14]. In Southeast Asia, the current CHBMS was translated and validated for an investigation of breast cancer screening behavior in many countries [14,15,16]. It was generally found to be a valid and reliable tool to measure the beliefs of multi-ethnic Southeast Asian women regarding breast cancer screening.

The CHBMS has been partially translated into Thai and used among the Thai population [17,18]; however, this tool has not been fully translated and tested for validity and reliability. The authors, therefore, sought to translate the CHBMS into Thai and test the translated version for validity and reliability. Moreover, the researchers added the ultrasound section, which is now commonly used in breast cancer screening, especially in younger Asian women who appear to have small and dense breasts compared to Western women. Unlike the widely accepted mammogram, there is still a lack of research data on its effectiveness of screening by ultrasound. However, it is normally used as it has been found to help detect breast cancer, especially in women with dense breast tissue. Although the NCCN guideline does not recommend routine ultrasound screening, it is mostly used to monitor treatment, especially for abnormalities on a mammogram or in young patients with breast abnormalities [17,18]. In some resource-limited areas, breast ultrasound has been proposed as a possible alternative for mammography in breast cancer screening because it is portable, less expensive than mammography, and versatile across a wider range of clinical applications. The use of ultrasound as an effective primary detection tool for breast cancer may be beneficial in low-resource settings where mammography is unavailable [19]. Furthermore, according to the findings of a multi-center randomized trial comparing ultrasound vs. mammography for screening breast cancer in high-risk Chinese women, ultrasound was superior to mammography for screening breast cancer in this group [19]. In Thailand, mammography is not available in most rural areas. Similarly, Thai women, like Chinese women, have smaller and denser breasts than Western women. Additionally, ultrasound yields less pain or discomfort than a mammogram, which is one of the main problems preventing women from breast cancer screening [7]. Currently, the technology of deep-learning-enabled clinical-decision-support systems for breast cancer diagnosis and classification on ultrasound images has been greatly developed. It is recognized as being effective in detecting breast cancer. Ultrasonography may become more prominent in the future for various breast-cancer-screening procedures [19]. However, this perspective about ultrasound was not included earlier, even in other Asian editions of CHBMS [19]. Therefore, this modified Thai version of the CHBMS can be used to assess both barrier and benefit perspectives comparing ultrasound and mammograms for breast cancer screening. In all, adding an ultrasound section to the questionnaire would make the questionnaire more complete in assessment.

We hypothesized that the translated version of the Thai CHBMS (T-CHBMS) and the modified questions for ultrasound would demonstrate appropriate validity and reliability.

## 2. Materials and Methods

A cross-sectional design was conducted for this study.

### 2.1. Participants

One hundred and thirty women were recruited from two health centers (Maharaj Nakorn Chiang Mai hospital—urban area and San Pa Tong hospital—rural area) in Chiang Mai, Thailand, from August 2021 to December 2021. The participants eligible for the study met the following criteria:Between the ages of 40 and 70 years (the recommended age for mammograms);No history of breast cancer or any other cancers;No pregnancy or breastfeeding.

Exclusion criteria included inability to communicate due to either language barrier or refusal to complete the questionnaires.

### 2.2. Development of the Modified Thai Version of CHBMS(MT-CHBMS)

The translation, adaptation, and cross-cultural content validation of an instrument for use in other cultures, languages, and countries require careful planning and adoption of comprehensive, rigorous, and the most established methodological approaches [15,16,20].

#### 2.2.1. Translation Process

The scale was translated using a forward back-translation technique.

##### Forward Translation

Translation from the original English version of the test into the Thai language was carried out by the authors with medical backgrounds and English proficiency. However, a bilingual person who was a lecturer at the Faculty of Humanities, Chiang Mai University did not have any medical involvement.

##### Synthesis of the Translations

The two versions were compared. Discussion and revision regarding the discrepancy were performed by the researcher team consisting of the investigators (PS, SJ, AS, TW, and NW), who were family medicine physicians, breast surgeons, and methodologists.

The language was checked grammatically and edited to be easy to understand and accurate in terms of medical terminology.

##### Backward Translation

A bilingual person from the Faculty of Arts, Media and Technology Modern Management and Technology, Chiang Mai University, who was not involved with forwarding translation translator and was unaware of Champion’s Health Belief Model Scale before, carried out backward translation. Then, both original and translated English versions were compared. Some minor discrepancy was found. The process was repeated with some discrepant items.

#### 2.2.2. Modification

The additional questions regarding ultrasound were created consistent with the mammograph questions. The research team examined the face and content validity for these newly added questionnaires and validity was confirmed by an expert panel. Content validity using the CVI index from 3 experts showed that the average Item-CVI was 1.00, which indicated excellent content validity. This new section was then back-translated to English. The final modified version was approved by the developer (Prof. Champion) (Figure 1).

#### 2.2.3. The Final MT-CHBMS

While the original CHBMS comprises 53 items for eight subscales, the MT-CHBMS comprises 64 questions for ten subscales: susceptibility (five items), seriousness (seven items), benefits of BSE (six items), barriers to BSE (six items), benefits of mammogram (six items), barriers to mammogram (five items), benefits of ultrasound (six items), barriers to ultrasound (five items), confidence (eleven items), and health motivation (seven items). The 11 items added to the original CHBMS included benefits of ultrasound (six items) and barriers to ultrasound (five items). Some examples of the questionnaire include “It is likely that I will get breast cancer”, “Having a mammogram will help me find lump early”, and “When I do breast self-examination, I feel good about myself”. The scales were measured with an ordinal scale using a five-point Likert type 1: “Strongly disagree”, to 5: “Strongly agree”. Each subscale can be used independently. In the case of overall assessment of the awareness of breast cancer and screening methods, the total score can be adopted but y questions concerning barriers must be reversed before summing up.

**Figure 1 healthcare-11-00128-f001:**
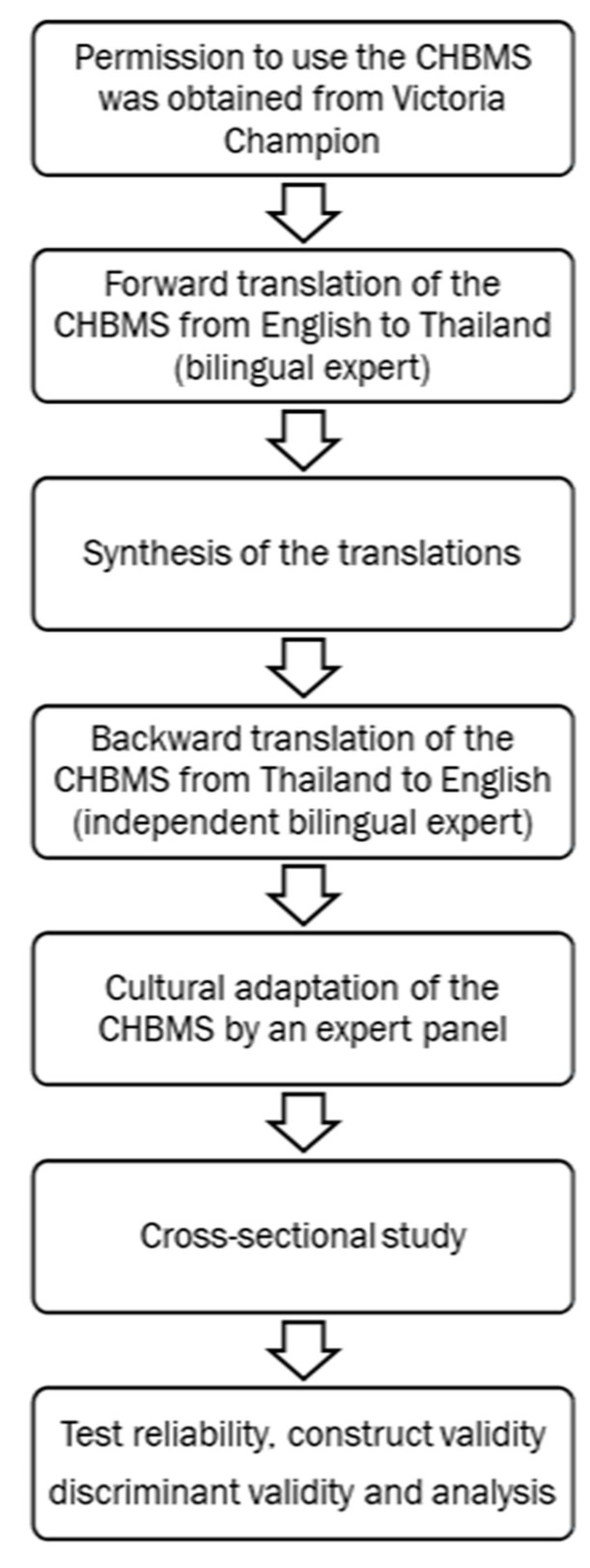
Cultural adaptation, translation, and validation of CHBMS.

### 2.3. Data Collection

Data were collected at an outpatient clinic through structured interviews by one of the investigators (PS), who had no role in patient care planning. All gave written informed consent before completing the questionnaires. The questionnaires included sociodemographic data, such as the respondent’s age, religion, marital status, education, healthcare insurance coverage, income, and residence area.

### 2.4. Statistical Analysis

Descriptive statistics were computed for the sociodemographic characteristics. The items for each subscale were examined for internal consistency using Cronbach’s alpha. The Cronbach alphas were calculated for all subscales and the full scale. The desired Cronbach alpha value is greater than 0.70 [21]. Construct validity was tested using confirmatory factor analysis (CFA). CFA was performed to examine the nature of and relations between latent constructs and to test how data were well-modelled with the designated construct. The CFA categorically tests a priori hypotheses about relations between observed and latent variables or factors. A model comparison was conducted between 8-factor (CHBM-T without two factors of ultrasound) and 10-factor models (CHBM-T with the factor of ultrasound (barrier and benefit). We used the following fit indexes: a CFI of 0.95, Tucker–Lewis index (TLI) of >0.9, a root-mean-square error of approximation (RMSEA) < 0.6, and chi-square/df < 3. Values as high as 0.08 indicated an acceptable fit [22,23,24]. CFA was carried out using the SPSS AMOS package version 18 [25].

## 3. Results

### 3.1. Sociodemographic Characteristics of Respondents

The average age of the sample was 52 years (SD = 7.28). Over 60% of women were single, Buddhists, and lived in Chiang Mai. Most participants’ educational level was high school to bachelor’s degree level. Almost all had health security (Table 1).

The item statistics of all 64 items were as follows. Most of the items (84.4%) fell from 1 to 5. The mean ranged from 2.38 to 4.58 (SD 0.560 to 1.214). Skewness values ranged from −1.487 to 0.364 (SE = 0.212), whereas kurtosis values ranged from −1.162 to 3.873 (SE = 0.422). All items’ skewness and kurtosis fell within the acceptable range (<±2 for skewness and <±7 for kurtosis) (see Appendix A).

Internal consistency WAS assessed by Cronbach’s alphas for each subscale to indicate that the items in the subscale measured the same construct. Table 2 shows the Cronbach’s alphas of the CHBMS and the MT-CHMBS. Overall, Cronbach’s alpha values were acceptable to excellent, ranging from 0.74 to 0.93 for the subscales. The mean and standard deviation of each score from the two studies were compared. In terms of the mean subscale score among the group, we found from ANOVA results that the benefits—BSE scores differed among ages; older participants scored higher than younger participants (*p* < 0.05). Participants who obtained a bachelor’s degree tended to score higher on barriers—BSE than participants who attained high school (*p* < 0.05) education. Women who had a higher income tended to score higher on barriers—BSE than participants with a lower income (*p* < 0.05). Participants who had the government or state enterprise health privilege (almost unlimited) scored a higher level on barriers—BSE than those who had the social security type of privilege (*p* < 0.01); the same was true for barriers—mammogram and barriers—ultrasound scores. Additionally, the barriers—mammogram scores were higher in universal coverage than in the social security group (*p* < 0.05). No difference was observed in different marital statuses.

### 3.2. Exploratory Factor Analysis (EFA)

To find the underlying structure of a large set of variables responded to by this sample, we conducted EFA using principal axis factoring. The Kaiser–Meyer–Olkin (KMO) measure was 0.74, indicating the data set was well-suited for factoring (KMO values less than 0.6 indicate the sampling is not adequate). In contrast, Bartlett’s test of sphericity was significant, suggesting a substantial correlation in the data. Using the eigenvalue of 1 for factor extraction, it initially yielded 15 components. The factor loadings of each item when ten factors were forced are shown in Table 3. Most items were loaded on the designated factor, except the items of the subscales barriers—BSE and barriers—mammogram that appeared to be combined into the same dimension. In contrast, the items from barriers—ultrasound seemed unable to form the designated dimension.

To further determine the possible factors from the EFA, Velicer’s minimum average partial (MAP) test was performed. It suggested to have 11 components as indicated by the smallest average square partial correlation and the fourth power partial correlation.

### 3.3. Confirmatiory Factor Analysis (CFA)

After the EFA, we further conducted confirmatory factor analysis (CFA) by specifying the number of factors required in the data and which measured variable was related to which latent variable. Two types of CFA were performed: the original 8-factor solution CFA (T-CHBMS) and the modified 10-factor solution CFA(MT-CHBMS). Table 4 shows the confirmatory factor analysis results of the CHBMS and MT-CHBMS. Each item had sufficient factor loadings (estimated coefficients) on the designated factor. All factor loading coefficients were significant (*p* < 0.001) and ranged from 0.413 to 1.029. The fit statistics were assessed to demonstrate how well the CFA model fitted the data.

The model fit statistics of the eight-factor solution of model T-CHBM were as follows: chi-square = 1701.977, df = 1274, chi-square/df = 1.336, TLI = 0.968, CFI = 0.970, and RMSEA (90% CI) = 0.051(0.044–0.057). For the model MT-CHBM: chi-square = 2488.868, df = 1879, chi-square/df = 1.324, TLI = 0.961, CFI= 0.964, and RMSEA (90% CI) = 0.050(0.045–0.055). Except for the motivation subscale, 21 pairs of error terms in each subscale of T-CHBMS and 23 pairs of error terms of MT-CHBMS were correlated. All these error terms suggested a high correlation between items and became the potential sources of the model misfit (see the Appendix A).

## 4. Discussion

This study examined the validity and reliability of the modified Thai version of the CHBMS, consisting of 10 subscales. The results have shown that it is a reliable and valid tool illustrated by an excellent content validity and confirmatory factor analysis. The findings have confirmed that each of the 10 subscales (susceptibility, seriousness, benefits of BSE, barriers to BSE, benefits of mammogram, barriers to mammogram, benefits of ultrasound, barriers to ultrasound, confidence, and health motivation) consists of the items significantly loaded on the designated subscales and can be used for assessment independently. Therefore, the Thai version of CHBMS and the newly modified MT-CHMBS are promising for measuring women’s beliefs about breast cancer and breast cancer screening in Thai women.

As the CHBMS has many factors, all subscales cannot be combined for the sum score and should be used separately. The new items regarding ultrasound seem to be consistent with scales for mammograms and may be combined. It is expected that the ultrasound section is related more to motivation than the self-examination part.

From exploratory factor analysis, it appeared that the respondents found it difficult to differentiate between types of barriers, evidenced by the fact that those barrier items were loaded on the same factor. Many items were shown to have cross-loadings, suggesting that a larger sample may be needed despite the fact that confirmatory factor analysis has confirmed that the 8-factor and 10-factor solution models were acceptable for this sample. In addition, we found that 23 pairs of error terms were suggested to be correlated in the model. This implies item duplication, resulting in the possibility of the scale being revised to have fewer items. For example, items S1 (It is extremely likely I will get breast cancer in the future) and S2 (I feel I will get breast cancer in the future) seem not to be able to be differentiated by the respondents. Likewise, the confidence subscales are I5 (I am able to find a breast lump which is the size of a quarter) and I6 (I am able to find a breast lump which is the size of a dime). We can see from the highly significant estimated coefficient of 0.858 (t = 38.118, *p* < 0.001) that one of these duplicating items should be deleted. Like in some versions, the CHBMS was shortened to increase compliance [20]. However, comparing the fitted items is problematic because, in the aforementioned study, only 19 items with the undifferentiated method were applied.

Despite the fact that those duplicated items may not cause damage to the whole sum score, redundant items, however, should be removed in order not to burden the respondents. Further investigation with a larger sample size should be warranted to confirm these problemed items, especially using the other method, such as item response theory.

In comparing the mean score of each subscale, it is surprising that the means of some subscales between the present study and Champion’s original study were close, even though there were many differences in terms of the culture, health system, and the time of data collection. For example, (susceptibility) 2.46 (SD = 0.98) and 2.54 (SD = 0.81); and (seriousness) 3.15 (SD = 0.81) and 3.25 (SD = 0.68) for the present study and Champion’s original study, respectively. For the newly added ultrasound items, the participants responded quite similarly to the mammogram items, suggesting the feasibility of these items for the modified Thai version of the CHBMS. We also found a difference in age, education, and health privilege scores. It is, however, rather difficult to tell whether it is from the actual difference or just due to item bias (differential item functioning). Therefore, a further step is encouraged to examine the possibility of the item biases.

It is fascinating to compare the findings in the Northern Thai population to the original scales for a diverse but mostly middle-class white community. It seems that the reliability of MT-CHBMS appears to have been higher than in the original version by Champion [9]. Cronbach’s alpha values of MT-CHBMS were acceptable to excellent, ranging from 0.74 to 0.93 for the subscales. All subscale values were superior to the original except barriers—mammograms.

Some studies in Asia have indicated that CHBMS has a good reliability. The Malaysia version of CHBMS recorded the Cronbach’s alpha values were acceptable to excellent, ranging from 0.77–0.93. This reliability is relatively similar to MT-CHBMS. The populations of the Malaysian version of CHBMS, on the other hand, were predominantly educated, married, and younger than our research population [14]; in addition, the Korean version of the CHBMS indicated a good reliability score, given the Cronbach’s alpha values ranged from 0.72–0.92. The majority of the population in the present study had technical college and bachelor’s degree education, had a low–middle income, were married, and were younger than the Malaysia and Thai version of CHBMS [13].

Thai women found that breast self-examination was a comparable benefit and barrier. In contrast, they believe much more in the benefits of mammograms than the barriers. It could be that such investigation is more accessible, and they may have more trust in the professionals than in their judgment on self-examination. As a matter of fact, mammogram is much better and shows a mortality benefit but not breast self-examination. Breast self-examination is used where there are not mammography facilities. Notably, the attitude towards ultrasound seems no different from that of the mammogram.

### 4.1. Clinical Implication and Future Research

This translated version of the CHBMS can be used to assess breast cancer knowledge and beliefs. Predictive validity by some relevant subscales may help the clinician develop a strategic plan to improve the targeted population’s awareness and practical examination. Modern test theory, such as item response theory, including the Rasch measurement model, should be further tested in addition to this classical test theory [26]. Moreover, a brief version of the CHBMS should be developed in future research.

### 4.2. Strength and Limitations

The study has demonstrated the construct validity of the modified version of CHBMS to which new items concerning ultrasound were added. This version should be appropriate for most Thai people with a dense breast mass. However, this study had some limitations—using participants in one specific geographic area of Northern Thailand. This hospital-based sample might limit the generalizability of the results to the general population. In addition, this study did not exclude participants with other breast masses and those with a family history of breast cancer that might affect that particular participant’s attitude toward breast cancer and screening methods. No external validation, e.g., concurrent validity, was conducted along with the construct validity. Test–retest reliability and predictive validity were not examined and should be included in future research.

## 5. Conclusions

The Thailand version of the modified CHBMS was estimated to be reliable and valid with Thailand women. This tool was fully translated and assessed for validity and reliability for the first time in Thailand. In addition, an adapted questionnaire for assessing the barrier and benefit of ultrasound was developed to help detect breast cancer, especially in women with dense breast tissue.

This study contributed a tool for assessing the perceived susceptibility, seriousness, health motivation, self-efficacy, benefits, and barriers of women regarding breast self-examination, mammograms, and ultrasound. Each of the ten subscales consists of items significantly loaded on the designated subscales and can be used for assessment independently. Primary care physicians, nurses, and other health care providers can use this tool to assess beliefs about breast cancer and breast cancer screening before making an appropriate health care plan. In addition, it may help the clinician develop a strategic plan to improve the targeted population’s awareness and create effective interventions suited to their beliefs. Furthermore, this tool can also be used as an assessment to measure the effects of breast cancer awareness and breast cancer screening activities and to reduce mortality from breast cancer with early detection among Thai women. Further investigation using item response theory, such as Rasch model, should be warranted, particularly when it comes to item reduction.

## Figures and Tables

**Table 1 healthcare-11-00128-t001:** Sociodemographic characteristics of the respondents (*n* = 130).

Characteristics	*n* (%) or Mean ± SD
Age (years), mean ± SD	52.33 ± 7.28
Marital status, *n* (%)	
Married	17 (13.08)
Single	80 (61.54)
Widow	15 (11.54)
Divorced	18 (13.85)
Education, *n* (%)	
- Unlettered	3 (2.31)
- Middle school	39 (30.00)
- High school/technical college	39 (30.00)
- Baccalaureate degree	37 (28.46)
- Master’s degree or higher	12 (9.23)
Monthly income, *n* (%)	
THB < 10,000	63 (48.46)
THB 10,000–14,999	16 (12.31)
THB 15,000–19,999	1 (0.77)
THB 20,000–24,999	6 (4.62)
THB > 25,000	44 (33.85)
THB < 10,000	
Health security, *n* (%)	
- Government or state enterprise officer	54 (41.54)
- Social security scheme	30 (23.08)
- Universal coverage scheme	44 (33.85)
- Private health insurance	0(0)
- Self-payment	2 (1.54)
SD = standard deviation	

**Table 2 healthcare-11-00128-t002:** Comparison of the Original Champion Health Beliefs Model with the modified champion health belief model—Thai version.

	Current Study (TM-CHBMS)		Champion’s Study (CHBMS)	
Subscale	Alpha	Mean (SD)	Alpha	Mean (SD)
Susceptibility	0.93	2.46 (0.98)	0.93	2.54 (0.81)
Seriousness	0.85	3.15 (0.81)	0.80	3.25 (0.68)
Benefits—BSE	0.88	4.05 (0.64)	0.80	3.88 (0.52)
Barriers—BSE	0.86	3.94 (0.65)	0.88	2.02 (0.60)
Confidence	0.90	4.10 (0.69)	0.88	3.31 (0.57)
Health motivation	0.85	3.81 (0.75)	0.83	3.78 (0.59)
Benefits—mammogram	0.93	25.36 (3.90)	0.79	23.86 (3.17)
Barriers—mammogram	0.74	19.05 (3.73)	0.75	11.02 (3.26)
Benefits—ultrasound	0.90	24.62 (4.12)	-	-
Barriers—ultrasound	0.79	19.05 (3.76)	-	-

T-CHBMS = Thai version of Champion’s Health Belief Model Scale, MT-CHBMS = modified Thai version of Champion’s Health Belief Model Scale, SD = standard deviation, BSE = breast self-examination.

**Table 3 healthcare-11-00128-t003:** Factor loadings of the MT-CHBMS using Principal Axis Factoring with Varimax with Kaiser Normalization.

Items	Factor									
	1	2	3	4	5	6	7	8	9	10
barb3	0.805			0.173			0.114		−0.116	
barb4	0.801				0.130			−0.131		
barb1	0.722		−0.115			−0.106	0.303			
barb5	0.721							−0.148		
bau2	0.673		0.187	0.176	0.187				0.216	
barb6	0.668				0.109		0.199	−0.101		0.147
barm2	0.664		0.187		0.139					
barm3	0.620		0.240				−0.120		0.314	
bau3	0.594		0.338		0.187				0.319	
bau4	0.534		0.234		0.194			−0.142	0.315	
barm1	0.532		0.200			0.110		−0.297	0.178	
barm4	0.448		0.177	0.119	0.108		−0.101		0.241	
barb2	0.359		−0.143		−0.126	−0.168	0.245	−0.284	0.290	
I5	0.123	0.776		0.134		0.126	−0.108		0.211	0.118
I6		0.775							0.211	
I11		0.760	0.158				0.195		−0.141	
I9		0.713	0.122	−0.143		−0.141	0.105			−0.133
I4		0.686			0.200		0.138			0.384
I10		0.685	0.153							
I8		0.683						−0.138	−0.147	
I3		0.678	−0.143		0.259		0.118			0.422
I7		0.638		−0.117		0.168				
bm3	0.182		0.816	0.165	0.199		0.131		0.119	
bm6	0.114		0.806	0.269	0.217			0.137		
bm2	0.141	0.104	0.767	0.272	0.175		0.169			0.114
bm4	0.168	0.165	0.740	0.229	0.255		0.221			
bm1	0.250		0.661	0.318	0.175		0.107			0.138
bm5	0.152	0.110	0.565	0.473	0.186		0.174			
beu1	0.134		0.288	0.822	0.247					0.181
beu3	0.129		0.314	0.796	0.175					
beu2		−0.105	0.177	0.788	0.148		0.219		0.117	
beu4	0.157		0.172	0.759	0.230	−0.118	0.114			
beu5			0.155	0.626	0.169		0.316		0.186	−0.247
beu6	0.113		0.382	0.477	0.206					
M4	0.152		0.258	0.225	0.845					
M3			0.215	0.181	0.799			0.143		
M2	0.166		0.108	0.270	0.736					
M5			0.152	0.116	0.709	−0.150				
M1	0.300		0.209	0.285	0.646	0.106		0.104		
M7			0.135	0.125	0.574	−0.175				
M6		0.158			0.386		0.254	−0.120		
S5						0.890				
S4						0.870		0.213	−0.118	
S2						0.848		0.174		
S3					−0.116	0.819		0.151		−0.151
S1						0.812		0.136	0.130	
be4	0.137	0.133	0.163	0.158			0.809			
be6	0.173	0.165	0.101	0.187	0.141		0.784	0.110		
be5		0.156	0.158	0.234	0.166		0.760	0.101		
be3	0.152		0.184	0.138		0.103	0.731			0.214
beb2	−0.130						0.683	0.113	0.275	0.388
se3	−0.157					0.104		0.820		
se2			0.110	0.102			0.104	0.733		
se1							0.226	0.705		
se4	−0.213					0.165	0.112	0.688		
se6								0.654		
se7	−0.180					0.161		0.517	−0.102	
se5				−0.102		0.314		0.455		0.146
barm5	0.248				0.103				0.558	
bau5	0.203						0.241	−0.145	0.531	0.137
bau1	0.487				−0.113		0.117	−0.151	0.490	−0.102
I2		0.530			0.111					0.531
I1	0.109	0.417		0.110	0.113					0.491
beb1			0.128				0.380	0.141		0.462

T-CHBMS = Thai version of Champion’s Health Belief Model Scale, MT-CHBMS = modified Thai version of Champion’s Health Belief Model Scale, S = item from susceptibility, se = item from seriousness, beb = item from benefit of breast self-examination, bm = item from benefit of mammogram, barb = item from barrier to breast self-examination, barm = item from barrier to mammogram, I = item from confidence, m = item from motivation, beu = item from benefit of ultrasound, and bau = item from barrier of ultrasound.

**Table 4 healthcare-11-00128-t004:** Confirmatory factor analysis results of the T-CHBMS and TM-CHBMS.

	T-CHBMS					MT-CHBMS			
	Estimate	S.E.	Est./S.E.	*p*-Value		Estimate	S.E.	Est./S.E.	*p*-Value
SUS	BY					BY			
S1	0.832	0.032	26.29	<0.001		0.832	0.032	26.094	<0.001
S2	0.873	0.021	40.605	<0.001		0.873	0.022	40.593	<0.001
S3	0.834	0.028	29.435	<0.001		0.833	0.028	29.273	<0.001
S4	0.903	0.022	41.627	<0.001		0.902	0.022	41.585	<0.001
S5	0.945	0.014	65.548	<0.001		0.946	0.014	66.01	<0.001
SERIOUSNESS	BY					BY			
SE1	0.726	0.043	16.873	<0.001		0.715	0.044	16.189	<0.001
SE2	0.752	0.051	14.614	<0.001		0.764	0.05	15.302	<0.001
SE3	0.818	0.038	21.287	<0.001		0.819	0.039	21.087	<0.001
SE4	0.806	0.035	23.32	<0.001		0.812	0.034	23.584	<0.001
SE5	0.532	0.06	8.83	<0.001		0.518	0.062	8.32	<0.001
SE6	0.647	0.057	11.277	<0.001		0.644	0.057	11.354	<0.001
SE7	0.594	0.060	9.858	<0.001		0.601	0.06	10.075	<0.001
BENEFIT of BSE	BY					BY			
BEB1	0.589	0.045	13.243	<0.001		0.566	0.049	11.664	<0.001
BEB2	0.629	0.051	12.393	<0.001		0.604	0.056	10.766	<0.001
BE3	0.830	0.031	26.849	<0.001		0.832	0.034	24.751	<0.001
BE4	0.915	0.021	44.273	<0.001		0.909	0.024	38.44	<0.001
BE5	0.944	0.020	46.91	<0.001		0.961	0.022	43.35	<0.001
BE6	0.958	0.025	37.853	<0.001		0.958	0.027	36.102	<0.001
BENEFIT of MG	BY					BY			
BM1	0.882	0.026	33.807	<0.001		0.899	0.024	37.499	<0.001
BM2	0.889	0.021	42.185	<0.001		0.893	0.022	41.473	<0.001
BM3	0.991	0.016	60.245	<0.001		0.977	0.017	56.99	<0.001
BM4	0.926	0.016	59.059	<0.001		0.925	0.016	58.723	<0.001
BM5	0.858	0.03	28.358	<0.001		0.906	0.024	37.257	<0.001
BM6	0.953	0.014	67.414	<0.001		0.953	0.014	68.123	<0.001
BARRIER to BSE	BY					BY			
BARB1	0.778	0.049	15.76	<0.001		0.772	0.05	15.55	<0.001
BARB2	0.455	0.067	6.847	<0.001		0.472	0.067	7.034	<0.001
BARB3	0.862	0.04	21.36	<0.001		0.858	0.04	21.227	<0.001
BARB4	0.910	0.034	26.871	<0.001		0.929	0.034	27.227	<0.001
BARB5	0.753	0.041	18.566	<0.001		0.740	0.043	17.262	<0.001
BARB6	0.836	0.041	20.621	<0.001		0.836	0.042	19.972	<0.001
BARRIER to MG	BY					BY			
BARM1	0.644	0.053	12.074	<0.001		0.653	0.052	12.506	<0.001
BARM2	0.891	0.04	22.452	<0.001		0.875	0.036	24.048	<0.001
BARM3	0.776	0.042	18.306	<0.001		0.756	0.04	19.05	<0.001
BARM4	0.606	0.061	9.993	<0.001		0.625	0.052	12.042	<0.001
BARM5	0.474	0.062	7.599	<0.001		0.515	0.056	9.192	<0.001
CONFIDENCE	BY					BY			
I1	0.679	0.056	12.154	<0.001		0.695	0.06	11.529	<0.001
I2	0.756	0.040	19.062	<0.001		0.752	0.042	18.022	<0.001
I3	0.905	0.030	29.666	<0.001		0.923	0.034	27.04	<0.001
I4	0.933	0.031	29.649	<0.001		0.953	0.035	26.976	<0.001
I5	0.739	0.045	16.376	<0.001		0.740	0.047	15.612	<0.001
I6	0.688	0.045	15.206	<0.001		0.681	0.047	14.359	<0.001
I7	0.489	0.058	8.474	<0.001		0.458	0.061	7.536	<0.001
I8	0.652	0.051	12.809	<0.001		0.641	0.054	11.879	<0.001
I9	0.670	0.047	14.255	<0.001		0.644	0.050	12.874	<0.001
I10	0.788	0.035	22.55	<0.001		0.783	0.037	20.892	<0.001
I11	0.776	0.042	18.354	<0.001		0.767	0.045	17.022	<0.001
MOTIVATION	BY					BY			
M1	0.88	0.034	25.653	<0.001		0.910	0.035	25.709	<0.001
M2	0.938	0.019	48.845	<0.001		0.941	0.02	46.929	<0.001
M3	0.933	0.017	56.135	<0.001		0.925	0.019	49.13	<0.001
M4	1.028	0.016	64.794	<0.001		1.029	0.016	63.697	<0.001
M5	0.779	0.041	19.054	<0.001		0.768	0.045	17.21	<0.001
M6	0.459	0.063	7.334	<0.001		0.413	0.067	6.138	<0.001
M7	0.719	0.044	16.288	<0.001		0.716	0.047	15.286	<0.001
					BENEFIT OF U	BY			
					BEU1	0.946	0.02	47.853	<0.001
					BEU2	0.863	0.028	30.303	<0.001
					BEU3	0.931	0.021	43.747	<0.001
					BEU4	0.922	0.023	39.612	<0.001
					BEU5	0.827	0.037	22.238	<0.001
					BEU6	0.809	0.036	22.385	<0.001
					BARRIER of U	BY			
					BAU1	0.622	0.047	13.269	<0.001
					BAU2	0.906	0.035	25.649	<0.001
					BAU3	0.880	0.027	32.853	<0.001
					BAU4	0.813	0.032	25.174	<0.001
					BAU5	0.494	0.064	7.741	<0.001

T-CHBMS = Thai version of Champion’s Health Belief Model Scale, MT-CHBMS = modified Thai version of Champion’s Health Belief Model Scale, SE = standard error, EST = estimated coefficient.

## Data Availability

The datasets used and/or analyzed during the current study are available from the corresponding author upon reasonable request.

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
