# Peer review of "Translation, Adaptation, and Validation of the Modified Thai Version of Champion’s Health Belief Model Scale (MT-CHBMS)"

_healthcare, 2022, doi:10.3390/healthcare11010128_

Round 1

Reviewer 1 Report (Previous Reviewer 1)

I thank the authors for responding to requests.

I think the article now meets the criteria to be published.

Author Response

21 December 2022

Dear Editor,

Thank you very much for providing us an opportunity to revise our manuscript. We have revise accordingly and response to those comments point-by-point as shown below,

Reviewer 1

I thank the authors for responding to requests.

I think the article now meets the criteria to be published.

Response: Thank you very much for your positive attitude.

Reviewer 2

  1. the authors need to change the abstract and focus more on the problem domain. before the paper's contributions, the author could precisely include the need of developing the proposed method
  2. Author response: We have revised the abstract as suggested. Please see below the revised one.

While breast cancer is the leading cause of cancer death among Thai women, breast self-examination (BSE), mammography, and ultrasound use are still infrequent. There is a need to assess women’s beliefs about Breast cancer and screening in different cultural settings. (Line 17) As a result, a tool to measure the beliefs that influence breast cancer screening practices is needed. The Champion's Health Belief Model Scale (CHBMS) is a valid and reliable tool for assessing individuals' attitudes toward breast cancer and screening methods, but it has not been validated in Thai women. The study aimed to translate and validate the CHBMS for breast self-examination and mammography among Thai women and to modify the original scale by adding ultrasound items for breast cancer screening. In addition, the purpose of this study was to create a modified Thai version of the CHBMS which could be used to better understand patients’ beliefs regarding breast cancer screening in Thailand, in order to develop practical and effective interventions suited to their beliefs.

  1. the novelty of this paper is not clear. the difference between the present work and previous works should be highlighted.
  2. Author response:

This research aims to develop a modified Thai version of the CHBMS for understanding Thai women's knowledge and beliefs about breast cancer screening, in order to create practical and effective interventions suited to their beliefs. (LIne 23)

“The author added question about ultrasound perspective in modified Thai version of the CHBMS, given that ultrasound is more accessible in Thailand and Other Asian countries”

This perspective about ultrasound was not included earlier even in other Asian editions of CHBMS. Therefore, this modified Thai version of the CHBMS can be used to assess both barriers and benefits perspectives comparing ultrasound and mammograms for breast cancer screening. (Line 122)

  1. the authors could better explain how "related works" is actually related to the current study. It is not clear to the reader how the menuscript is similar to or differs from these related works.
  2. Author response: already adding in question 2

  1. some recent work should be added, such as:
  2. Author response:

We will add this recent work in the introduction, thank you for your advice.

..Also, ultrasound yields less pain or discomfort than a mammogram, which is one of the main problems preventing women from breast cancer screening[6]. Currently, the technology of deep-learning-enabled clinical decision support systems for breast cancer diagnosis and classification on Ultrasound Images has been greatly developed. It is recognized as being effective in detecting breast cancer. Ultrasonography may become more prominent in the future for various breast cancer screening procedures[26-28].

https://doi.org/10.3390/biology11030439

https://doi.org/10.32604/cmc.2022.022322

https://doi.org/10.1155/2022/3714422

  1. Result needs more explanations. Additional analysis is required at each experiment to show the main purpose.
  2. Author response: we have added more explanations for each part of the results (lines 232, 249, 268, 273)
  3. The authors have used some mathematical notations. make sure that all the parameters are described, and also check the mathematical notations
  4. Author response: we have already checked them. Thank you.
  5. How did the authors apply the augmentations technique
  6. Author response:

“The authors think that the augmentations technique will be useful primarily in the field of breast cancer diagnosis technic that mostly refer to biomedical science research. in addition, the augmentations technique will be beneficial in the helping work of radiologists and HNB surgeons. It can help decrease errors from diagnosis especially false positives in mammograms. However, this research will focus on beliefs and attitudes about breast cancer screening. So, the design of this research will be more likely psychology research. Moreover, the deep transfer learning-based breast cancer detection will use a high technology computer that inadequate resources in developing countries like Thailand for widely screening breast cancer. In the future, if the augmentation technique were developed in the application or lower cost of technology, the study belief of patients about AI with ultrasound for breast cancer screening may be useful.”

  1. the manuscript is well-organized and properly formatted. the authors are suggested to have the paper revised to improve the language.
  2. Author response: Already done by native language speaker

  1. the conclusion should indicate the experimental evaluation's implication and include some obtained values to clearly point out the superiority
  2. Author response:

The Thailand version of the modified CHBMS was estimated to be reliable and valid with Thailand women.

(Line 384) This tool was fully translated and assessed for validity and reliability for the first time in Thailand. In addition, an adapted questionnaire for assessing the barrier and benefit of ultrasound is developed to help detect breast cancer, especially in women with dense breast tissue.

This study contributed a tool for assessing the perceived susceptibility, seriousness, health motivation, self-efficacy, benefits and barriers of women regarding breast self-examination, mammograms, and ultrasound. Each of the ten subscales consists of items significantly loaded on the designated subscales and can be used for assessment independently. Primary care physicians, nurses, and other health care providers can use this tool to assess beliefs about breast cancer and breast cancer screening before making an appropriate health care plan. In addition, it may help the clinician develop a strategic plan to improve the targeted population’s awareness and create effective interventions suited to their beliefs.

Reviewer3

  1. Thanks for the response. However, I still suggested more Participants and Criteria should be introduced to increase the completion of the results
  2. Author response:

Thank you for your recommendation; we appreciate your kindness and excellent guidance.

However, as our study has already concluded the interview participant method and we see your suggestion as our research limitation. Therefore, we have added what the reviewer has suggested in the limitation.

We hope our response would sufficiently meet the reviewer expectation. We are looking forward to hearing from you soon.

Best,

SJ and TW

Reviewer 2 Report (Previous Reviewer 2)

·         The authors need to change the abstract and focus more on the problem domain. Before the paper's contributions, the author could precisely include the need of developing the proposed method.

·         The novelty of this paper is not clear. The difference between the present work and previous Works should be highlighted.

·         The authors could better explain how “Related works” is actually related to the current study. It is not clear to the reader how the manuscript is similar to or differs from these related works.

·          Some recent works should be added, such as: https://doi.org/10.3390/biology11030439, https://doi.org/10.32604/cmc.2022.022322, https://doi.org/10.1155/2022/3714422

·         Results need more explanations. Additional analysis is required at each experiment to show the main purpose.

·         The authors have used some mathematical notations. Make sure that all the parameters are described. And also check the mathematical notations.

·         How did the authors apply the Augmentation technique?

·         The manuscript is well-organized and properly formatted. The authors are suggested to have the paper revised to improve the language.

·          The conclusion should indicate the experimental evaluation's implications and include some obtained values to clearly point out the superiority.

Author Response

21 December 2022

Dear Editor,

Thank you very much for providing us an opportunity to revise our manuscript. We have revise accordingly and response to those comments point-by-point as shown below,

Reviewer 1

I thank the authors for responding to requests.

I think the article now meets the criteria to be published.

Response: Thank you very much for your positive attitude.

Reviewer 2

  1. the authors need to change the abstract and focus more on the problem domain. before the paper's contributions, the author could precisely include the need of developing the proposed method
  2. Author response: We have revised the abstract as suggested. Please see below the revised one.

While breast cancer is the leading cause of cancer death among Thai women, breast self-examination (BSE), mammography, and ultrasound use are still infrequent. There is a need to assess women’s beliefs about Breast cancer and screening in different cultural settings. (Line 17) As a result, a tool to measure the beliefs that influence breast cancer screening practices is needed. The Champion's Health Belief Model Scale (CHBMS) is a valid and reliable tool for assessing individuals' attitudes toward breast cancer and screening methods, but it has not been validated in Thai women. The study aimed to translate and validate the CHBMS for breast self-examination and mammography among Thai women and to modify the original scale by adding ultrasound items for breast cancer screening. In addition, the purpose of this study was to create a modified Thai version of the CHBMS which could be used to better understand patients’ beliefs regarding breast cancer screening in Thailand, in order to develop practical and effective interventions suited to their beliefs.

  1. the novelty of this paper is not clear. the difference between the present work and previous works should be highlighted.
  2. Author response:

This research aims to develop a modified Thai version of the CHBMS for understanding Thai women's knowledge and beliefs about breast cancer screening, in order to create practical and effective interventions suited to their beliefs. (LIne 23)

“The author added question about ultrasound perspective in modified Thai version of the CHBMS, given that ultrasound is more accessible in Thailand and Other Asian countries”

This perspective about ultrasound was not included earlier even in other Asian editions of CHBMS. Therefore, this modified Thai version of the CHBMS can be used to assess both barriers and benefits perspectives comparing ultrasound and mammograms for breast cancer screening. (Line 122)

  1. the authors could better explain how "related works" is actually related to the current study. It is not clear to the reader how the menuscript is similar to or differs from these related works.
  2. Author response: already adding in question 2

  1. some recent work should be added, such as:
  2. Author response:

We will add this recent work in the introduction, thank you for your advice.

..Also, ultrasound yields less pain or discomfort than a mammogram, which is one of the main problems preventing women from breast cancer screening[6]. Currently, the technology of deep-learning-enabled clinical decision support systems for breast cancer diagnosis and classification on Ultrasound Images has been greatly developed. It is recognized as being effective in detecting breast cancer. Ultrasonography may become more prominent in the future for various breast cancer screening procedures[26-28].

https://doi.org/10.3390/biology11030439

https://doi.org/10.32604/cmc.2022.022322

https://doi.org/10.1155/2022/3714422

  1. Result needs more explanations. Additional analysis is required at each experiment to show the main purpose.
  2. Author response: we have added more explanations for each part of the results (lines 232, 249, 268, 273)
  3. The authors have used some mathematical notations. make sure that all the parameters are described, and also check the mathematical notations
  4. Author response: we have already checked them. Thank you.
  5. How did the authors apply the augmentations technique
  6. Author response:

“The authors think that the augmentations technique will be useful primarily in the field of breast cancer diagnosis technic that mostly refer to biomedical science research. in addition, the augmentations technique will be beneficial in the helping work of radiologists and HNB surgeons. It can help decrease errors from diagnosis especially false positives in mammograms. However, this research will focus on beliefs and attitudes about breast cancer screening. So, the design of this research will be more likely psychology research. Moreover, the deep transfer learning-based breast cancer detection will use a high technology computer that inadequate resources in developing countries like Thailand for widely screening breast cancer. In the future, if the augmentation technique were developed in the application or lower cost of technology, the study belief of patients about AI with ultrasound for breast cancer screening may be useful.”

  1. the manuscript is well-organized and properly formatted. the authors are suggested to have the paper revised to improve the language.
  2. Author response: Already done by native language speaker

  1. the conclusion should indicate the experimental evaluation's implication and include some obtained values to clearly point out the superiority
  2. Author response:

The Thailand version of the modified CHBMS was estimated to be reliable and valid with Thailand women.

(Line 384) This tool was fully translated and assessed for validity and reliability for the first time in Thailand. In addition, an adapted questionnaire for assessing the barrier and benefit of ultrasound is developed to help detect breast cancer, especially in women with dense breast tissue.

This study contributed a tool for assessing the perceived susceptibility, seriousness, health motivation, self-efficacy, benefits and barriers of women regarding breast self-examination, mammograms, and ultrasound. Each of the ten subscales consists of items significantly loaded on the designated subscales and can be used for assessment independently. Primary care physicians, nurses, and other health care providers can use this tool to assess beliefs about breast cancer and breast cancer screening before making an appropriate health care plan. In addition, it may help the clinician develop a strategic plan to improve the targeted population’s awareness and create effective interventions suited to their beliefs.

Reviewer3

  1. Thanks for the response. However, I still suggested more Participants and Criteria should be introduced to increase the completion of the results
  2. Author response:

Thank you for your recommendation; we appreciate your kindness and excellent guidance.

However, as our study has already concluded the interview participant method and we see your suggestion as our research limitation. Therefore, we have added what the reviewer has suggested in the limitation.

We hope our response would sufficiently meet the reviewer expectation. We are looking forward to hearing from you soon.

Best,

SJ and TW

Reviewer 3 Report (Previous Reviewer 3)

Thanks for the response. However, I still suggested more Participants and Criteria should be introduced to increase the completion of the results. 

Author Response

21 December 2022

Dear Editor,

Thank you very much for providing us an opportunity to revise our manuscript. We have revise accordingly and response to those comments point-by-point as shown below,

Reviewer 1

I thank the authors for responding to requests.

I think the article now meets the criteria to be published.

Response: Thank you very much for your positive attitude.

Reviewer 2

  1. the authors need to change the abstract and focus more on the problem domain. before the paper's contributions, the author could precisely include the need of developing the proposed method
  2. Author response: We have revised the abstract as suggested. Please see below the revised one.

While breast cancer is the leading cause of cancer death among Thai women, breast self-examination (BSE), mammography, and ultrasound use are still infrequent. There is a need to assess women’s beliefs about Breast cancer and screening in different cultural settings. (Line 17) As a result, a tool to measure the beliefs that influence breast cancer screening practices is needed. The Champion's Health Belief Model Scale (CHBMS) is a valid and reliable tool for assessing individuals' attitudes toward breast cancer and screening methods, but it has not been validated in Thai women. The study aimed to translate and validate the CHBMS for breast self-examination and mammography among Thai women and to modify the original scale by adding ultrasound items for breast cancer screening. In addition, the purpose of this study was to create a modified Thai version of the CHBMS which could be used to better understand patients’ beliefs regarding breast cancer screening in Thailand, in order to develop practical and effective interventions suited to their beliefs.

  1. the novelty of this paper is not clear. the difference between the present work and previous works should be highlighted.
  2. Author response:

This research aims to develop a modified Thai version of the CHBMS for understanding Thai women's knowledge and beliefs about breast cancer screening, in order to create practical and effective interventions suited to their beliefs. (LIne 23)

“The author added question about ultrasound perspective in modified Thai version of the CHBMS, given that ultrasound is more accessible in Thailand and Other Asian countries”

This perspective about ultrasound was not included earlier even in other Asian editions of CHBMS. Therefore, this modified Thai version of the CHBMS can be used to assess both barriers and benefits perspectives comparing ultrasound and mammograms for breast cancer screening. (Line 122)

  1. the authors could better explain how "related works" is actually related to the current study. It is not clear to the reader how the menuscript is similar to or differs from these related works.
  2. Author response: already adding in question 2

  1. some recent work should be added, such as:
  2. Author response:

We will add this recent work in the introduction, thank you for your advice.

..Also, ultrasound yields less pain or discomfort than a mammogram, which is one of the main problems preventing women from breast cancer screening[6]. Currently, the technology of deep-learning-enabled clinical decision support systems for breast cancer diagnosis and classification on Ultrasound Images has been greatly developed. It is recognized as being effective in detecting breast cancer. Ultrasonography may become more prominent in the future for various breast cancer screening procedures[26-28].

https://doi.org/10.3390/biology11030439

https://doi.org/10.32604/cmc.2022.022322

https://doi.org/10.1155/2022/3714422

  1. Result needs more explanations. Additional analysis is required at each experiment to show the main purpose.
  2. Author response: we have added more explanations for each part of the results (lines 232, 249, 268, 273)
  3. The authors have used some mathematical notations. make sure that all the parameters are described, and also check the mathematical notations
  4. Author response: we have already checked them. Thank you.
  5. How did the authors apply the augmentations technique
  6. Author response:

“The authors think that the augmentations technique will be useful primarily in the field of breast cancer diagnosis technic that mostly refer to biomedical science research. in addition, the augmentations technique will be beneficial in the helping work of radiologists and HNB surgeons. It can help decrease errors from diagnosis especially false positives in mammograms. However, this research will focus on beliefs and attitudes about breast cancer screening. So, the design of this research will be more likely psychology research. Moreover, the deep transfer learning-based breast cancer detection will use a high technology computer that inadequate resources in developing countries like Thailand for widely screening breast cancer. In the future, if the augmentation technique were developed in the application or lower cost of technology, the study belief of patients about AI with ultrasound for breast cancer screening may be useful.”

  1. the manuscript is well-organized and properly formatted. the authors are suggested to have the paper revised to improve the language.
  2. Author response: Already done by native language speaker

  1. the conclusion should indicate the experimental evaluation's implication and include some obtained values to clearly point out the superiority
  2. Author response:

The Thailand version of the modified CHBMS was estimated to be reliable and valid with Thailand women.

(Line 384) This tool was fully translated and assessed for validity and reliability for the first time in Thailand. In addition, an adapted questionnaire for assessing the barrier and benefit of ultrasound is developed to help detect breast cancer, especially in women with dense breast tissue.

This study contributed a tool for assessing the perceived susceptibility, seriousness, health motivation, self-efficacy, benefits and barriers of women regarding breast self-examination, mammograms, and ultrasound. Each of the ten subscales consists of items significantly loaded on the designated subscales and can be used for assessment independently. Primary care physicians, nurses, and other health care providers can use this tool to assess beliefs about breast cancer and breast cancer screening before making an appropriate health care plan. In addition, it may help the clinician develop a strategic plan to improve the targeted population’s awareness and create effective interventions suited to their beliefs.

Reviewer3

  1. Thanks for the response. However, I still suggested more Participants and Criteria should be introduced to increase the completion of the results
  2. Author response:

Thank you for your recommendation; we appreciate your kindness and excellent guidance.

However, as our study has already concluded the interview participant method and we see your suggestion as our research limitation. Therefore, we have added what the reviewer has suggested in the limitation.

We hope our response would sufficiently meet the reviewer expectation. We are looking forward to hearing from you soon.

Best,

SJ and TW

Round 2

Reviewer 2 Report (Previous Reviewer 2)

The authors answered all comments 

This manuscript is a resubmission of an earlier submission. The following is a list of the peer review reports and author responses from that submission.

Round 1

Reviewer 1 Report

The article investigates the Champion's Health Belief Model Scale (CHBMS) as a tool to assess the individuals' attitudes toward breast cancer and screening method in Thailand.

The interesting interesting topic should be better developed and presented.

Here some suggestion:

introduction: line 48 check punctuation after reference 5

So if I understand correctly from the introduction of your paper, in Thailand,  is there currently no mammography screening? this should be better explained in the text. In Thailand do you only offer mammography to women after the age of 70? it is not specified whether there is, or is not, a national screening program, in your country.

Line 67: therefore….please rephrase this sentence that is too contorted

Line 69: please add the Champion's Health Belief Model Scale (CHBMS) story. When did it originate? where is it used?

Line 81: So is breast ultrasound in Asia commonly used in mammography screening or was it the result of an isolated study? how does this link with the fact that you mentioned earlier that screening in Asia is less felt? how do you solve the problem of more resources needed to perform breast ultrasound in a screening setting? you need to better explain what the current standardized screening pathway is in your country

Matherial and methods

you should specify whether this study has been notified to the ethics committee.

If so, you should produce a proposal acceptance number from the ethics committee

Discussion:  The results have shown that it is a reliable and valid 233 tool illustrated by an excellent content validity and confirmatory factor analysis” you should summarize in the discussion what are the statistically significant findings that have clinical impact.

There are too many abbreviations in the discussion that make it difficult to follow the text.

A facsimile of the questionnaire should be presented in the text

“line 261”: “it is surprising that the means between 261 the present study and the Champion’s original study were close”: to use this kind of statement one would have to perform a comparison of the results obtained in numerical terms.

Conclusion: you mention the rash model: you should describe it better in the text and add appropriate bibliographic citations.

Reviewer 2 Report

·         The authors need to change the abstract and focus more on the problem domain. Before the paper's contributions, the author could precisely include the need to develop the proposed method.

·         The novelty of this paper is not clear. The difference between the present work and previous Works should be highlighted.

·         The authors could better explain how “Related works” is actually related to the current study. It is not clear to the reader how the manuscript is similar to or differs from these related works.

·           Some recent works should be added

·         cture of the proposed methods

·         The authors have used some mathematical notations. Make sure that all the parameters are described. And also check the mathematical notations.

·         The manuscript is well-organized and properly formatted. The authors are suggested to have the paper revised to improve the language

Reviewer 3 Report

The paper is very well written and proposes a new method of increasing breast cancer early detection. The authors translated the Champion's Health Belief Model Scale (CHBMS) tool into Thai, which helps for measuring individuals’ attitudes toward breast cancer and screening methods in Thailand.

However, some issues should be addressed before its publication.

1. The Participant limitation

In-Page 2 line 92, the authors demonstrated that “One hundred and thirty women were recruited from two health centers in Chiang-Mai, Thailand”. It is well-known that Chiang-Mai is the largest city in northern Thailand and the second largest city in Thailand. The participants are all from the developed city, thus, the obtained conclusions are limited. How about the results from backward rural areas?

Furthermore, In Page 5 Table 1, the authors studied the sociodemographic characteristics of the respondents, including Age, Marital status, Education, Monthly income, and Health security. Here, there is a significant difference between city and rural in Education, Monthly income, and Health security. For example, the number of Self-payment participants in Table 1 is only 2. However, most women from rural areas have no health insurance, thus the number of Self-payment people will increase.

2. Medical history limitation

In-Page 3 line 96, the authors said the participants eligible for the study met the criteria: “no history of breast cancer or any other cancers”. This medical history is limited. How about other breast diseases? Such as Fibrocystic breast condition, cysts, fibroadenomas, mastitis, nipple discharge, and calcifications. Women with or without any breast common diseases have a total difference awareness of breast cancer early detection.

Besides, how about the family medical history? For example, if the mother in a family has breast cancer, the daughters, and sisters will have more cautious attitudes toward early detection.

So, I suggested more Participants and Criteria should be introduced to increase the completion of the results.

3. Text editing

The words in Figure 1 are not clear, could you please change to a high-definition picture?

The line widths in Table 1 are not uniform.